# Design of Three-Phase V-Shaped Interior Permanent Magnet Synchronous Motor for Air Conditioning Compressor of Electric Vehicle

Hojin Jeong  and Jeihoon Baek *

Electrical &Electronics and Communication Engineering Department, Koreatech University, Cheonan-si 31253, Korea; husik5864@koreatech.ac.kr
* Correspondence: jhbaek@koreatech.ac.kr; Tel.: +82-41-560-1258

**Abstract:** Air conditioning system of electric vehicles has new change as the internal combustion engine is being replaced with electrified AC motor. With large amount of batteries installed at the bottom of frame, the conventional compressor, which is belt-driven, can be removed, and another AC motor can play the role for air conditioning in electric vehicles. From this change, the system efficiency would be improved since it is possible to control the electrified compressor independently from traction system in contrast with the belt-driven compressor. As a result, by applying the electrified compressor for air conditioning system, the whole system can achieve better efficiency and longer driving distance, which is most important in electric vehicles. In this paper, 3-phase interior permanent magnet synchronous motor (IPMSM) was designed using lumped-parameter model and finite element method.

**Keywords:** IPMSM; compressor; V-shaped PM; electric vehicle; air conditioner

## 1. Introduction

The conventional compressor installed with internal combustion engine for air conditioner is driven by being connected with a timing belt, which is so called belt-driven compressor. However, this sort of compressor has a disadvantage in efficiency since its operating point depends on the speed of crank shaft of engine, and thus, the compressor is not allowed to be independently driven at desirable operating points where high efficiency is ensured. As electric vehicles get to be supplied broadly, innovation in the involved subsystem is needed to save energy. Especially in the air conditioning system, as it is possible for subsystems to share battery pack as power source with power train, the adoption of an electrified motor and inverter is considered so as to utilize the independent operation for high efficiency.

Energy consumed by air conditioner would be significant. Since one of the most important factors in electric vehicles is the extension of driving distance, high efficiency and power density (per unit weight or unit volume) of subsystems belong to the most important factors. Conventional actuators used for air conditioner are in the form of induction machines and scalar control method for low cost and simple implementation. Because, in the common environments of air conditioner system, induction machine is usually fixed somewhere in factories or buildings, power density is not considered as important as in mobile machine-like electric vehicles. Actually, the type of motor that achieves high efficiency and power density is interior permanent magnet synchronous motor (IPMSM) [1,2].

By adopting IPMSM with rare earth permanent magnet (PM) such as NdFeB in place of induction machine (IM), it is possible to meet and improve the most requirements in efficiency, power density and power factor [3–5]. Throughout the literature, the efficiency of IPMSM ranged from 93 to 95%

while IM achieved from 85 to 87% in the same size [3]. In another literature, power factor of PMSM, 0.95, was significantly higher than the one of IM, 0.89, and power density of PMSM was improved by 17.3% [4]. However, in addition to the performances mentioned above, there are two more parameters to be suppressed: torque ripple, electromagnetic interference (EMI) noise and cost. The inherent cause of torque ripple is originated from magnetomotive force (MMF) subharmonics of a certain winding method in stator and no-load flux density on air gap formed by interior PM and the shape of magnet cavity in rotor. Moreover, the external causes exist like switching harmonics and imperfect fabrication. These reasons lead to fluctuation of speed in open-loop or closed loop system whose sampling time is long and, as a result, uncomfortable acoustic noise and mechanical vibration inside vehicles may be generated [6–8]. Thus, in the various industry applications, torque ripple should be suppressed for low acoustic noise via dedicated control methods or initial system design [9–12]. Likewise, EMI noise is caused in the almost same reasons, switching harmonics and harmonics in control input in the control system [13]. Electromagnetic compatibility requirements in the automotive sector are most stringent; inherent causes must be excluded as much as possible in motor design. Now that raw material cost is considered here, the most expensive material is rare earth PM. In case of mass production, such as automotive, raw material cost is a sensitive issue. Therefore, design optimization is needed, where PM volume should be minimized ensuring enough performance [14].

That is, the goal of machine design for air conditioning compressor in electric vehicles is to optimize for maximized efficiency and power density, while at the same time, suppressing torque ripple and PM volume [15]. IPMSM is comprised of stator and rotor that has interior magnet cavity and inserted PM there. For blushless AC machine where sinusoidal back-electromotive force (back-EMF) is desired, stator is required to generate as high main harmonics as possible and as low sub-harmonics as possible in MMF distribution regardless of rotor type. It is clearly summarized in form of winding factor [16–18]. On the other hand, rotor design determines the performances such as d, q-axis inductances and no-load flux linkage corresponding to shape of cavities, PM volume, and configuration of PM. Especially, raw material cost is concentrated on rotor in which PM is contained. Thus, design of rotor has more effect on improvements in torque ripple, cost and efficiency. Literatures [19–21] investigated the trends of performances with respect to various rotor types at given stators: distributed winding and concentrated winding each. This paper investigates the characteristic trends when gradually changing rotor design of V-shaped IPMSM, which has the moderate performances in copper loss, torque ripple, cost, and efficiency [20].

Above all, it is envisaged in basis of lumped parameter model (LPM). Vagati suggests the analysis of synchronous reluctance machine (SynRM) using d and q-axis magnetic equivalent circuit (MEC) as LPM [22]. Furthermore, E. Lovelace suggests the optimization process of three-phase IPMSM with two layered ferrite PM using LPM and differential evolutionary strategy (DES) in his dissertation [23]. In 2014, J. Baek designed five-phase PM assisted SynRM (PMaSynRM) with rare earth PM using the optimization process [24,25]. Afterwards, S. Sudheer designed and compared five-phase PMaSynRM whose external rotor is constructed with single layered cavity to three-phase PMaSynRM [26,27]. The optimization with LPM and DES has been implemented and used many times for PMaSynRM and IPMSM with ferrite PM, however, not for V-shaped IPMSM especially with large amount of rare earth PM. Since Vagati's LPMs on d, q-axis MEC are considered for multi-layered SynRM without any magnet, it does not consider the existence of rare earth PM strong enough to saturate the cores at initial state (no-load state) [22,28]. This initial saturation would lower the accuracy of LPM. Therefore, the deductions based on LPM must be validated in comparison with the results from finite element analysis (FEA) and performance test on a fabricated motor.

Therefore, this paper investigates the characteristic trends according to rotor design in the following process. First, the cavities in rotor are V-shaped because V-shaped cavity is superior to flat or spoke type in torque ripple, copper loss and cost according to the literature [19,21]. Single layered IPMSM will be designed with five different shapes of cavity which gradually grow the included angle in V-shape. Moreover, Section 2 introduces d, q-axis MEC constructed in LPM and envisages how

parameters will be affected with respect to each case of the shapes. In Section 3, FEA results will be represented to see the agreement between LPM and FEA. Finally, experimental results will be present.

## 2. Lumped Parameter Model of V-Shaped IPMSM

LPM is an MEC for estimating d and q-axis inductances and no-load flux linkage from interior PM, which are required to calculate torque performance at certain current values in Equation (1). Torque equation is the sum of magnetic torque and reluctance torque. Magnetic torque is the first term of Equation (1), which is the product of PM flux linkage and q-axis current, while reluctance torque is the product of difference in d, q-axis inductances and each current, $I_d$ and $I_q$.

$$T_{em} = \frac{3}{2}p\left[\lambda_{PM}I_q + \left(L_d - L_q\right)I_dI_q\right]$$
(1)

where $T_{em}$: generated torque of IPMSM, $p$: number of pole pair, $I_d$: d-axis current, $I_q$: q-axis current, $L_d$: d-axis inductance, $L_q$: q-axis inductance.

### 2.1. PM Flux Linkage

Figure 1 describes the MEC to be solved for the estimation of PM flux linkage which contributes to magnetic torque. This MEC only considers linear reluctances of air gap and flux barrier of magnet cavity with flux sources from PM and a bridge that is a narrow part in rotor between air gap and magnet cavity. To estimate PM flux linkage, flux density on air gap must be calculated from MEC by finding $\phi_g$ flowing on $r_g$ using superposition principle. The individual elements can be calculated as below.

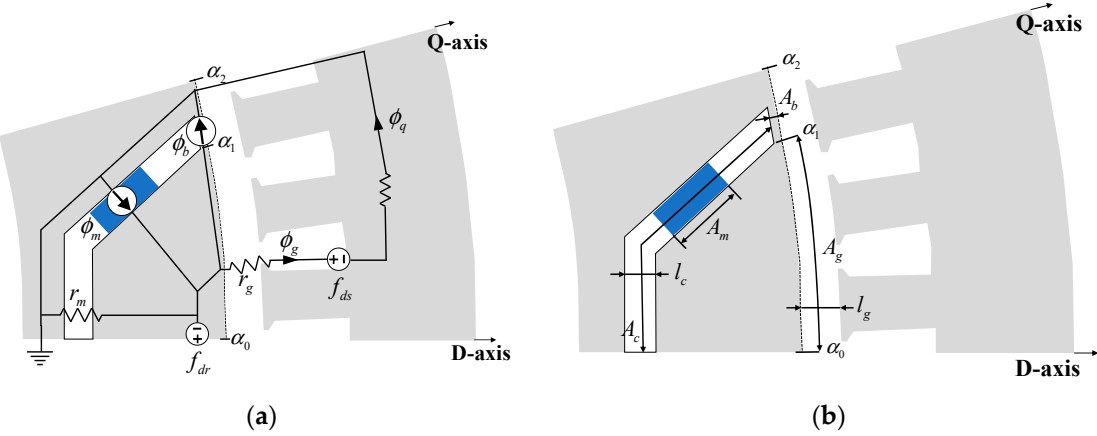

(**a**)　　　　　　　　　　　　　　　　(**b**)

**Figure 1.** Magnetic equivalent circuit (MEC) for d-axis permanent magnet (PM) flux linkage (**a**) d-axis linear MEC for PM flux linkage. (**b**) Geometric definition for reluctances.

$$r_m = \frac{l_c}{\mu_0 A_c}, \ r_g = \frac{l_g}{\mu_0 A_g}$$
(2)

$$\phi_b = B_{sat}A_b, \ \phi_m = B_rA_m$$
(3)

$$B_g = \frac{r_m}{r_m + r_g}\frac{\phi_m - \phi_b}{A_g}$$
(4)

$$B_1 = \frac{4}{\pi}B_g(\sin(\alpha_0) - \sin(\alpha_1))$$
(5)

$$\lambda_{PM} = \frac{\sqrt{2}rlB_1N_ak_w}{p}$$
(6)

where $r_{m,g}$: linear reluctances of magnet cavity and airgap, [H$^{-1}$], $l_{c,g}$: length of magnet cavity and airgap, [m], $A_{c,g}$: area of magnet cavity and airgap, [m$^2$], $\phi_b$: leakage flux passing through bridge, [Wb], $\phi_m$: total remanent flux from PM, [Wb], $A_b$: area of magnetic path on bridge, [m$^2$], $A_m$: area of PM contacted on core, [m$^2$], $B_{sat}$: flux density of the saturated bridge, [T], $B_m$: remanent flux density of PM, [T], $B_g$: flux density on airgap, [T], $B_1$: fundamental amplitude of flux density distributed on airgap, [T], $\alpha_{0,1}$: electrical angular positions depicted in Figure 1, [rad], $\lambda_{PM}$: flux linkage at no-load, [Wb], $r$: inner radius of stator, [m], $l$: stack length, [m], $N_a$: serial turns per phase, $k_w$: winding factor. Air gap reluctance, $r_g$, is distributed from $\alpha_0$ to $\alpha_1$. $\phi_b$ is evaluated on the assumption that a bridge is fully saturated at $B_{sat}$ by PM flux. Flux density on air gap from $\alpha_0$ to $\alpha_1$ is calculated from $\phi_g$ and $A_g$ as in Equation (4), and flux density from $\alpha_1$ to $\alpha_2$ is assumed to be 0 according to the results from FEA. Thus, flux density distributed on air gap at no load will be of rectangular shape. The amplitude of fundamental component of flux density is calculated with Equation (5). Finally, PM flux linkage is obtained via Equation (6).

Analyzing the circuit and formulations, as the included angle between two wings of magnet cavity becomes narrower, the area of magnet cavity becomes larger. It is expected that its reluctance $r_m$ in Equation (1) be diminished and more PM be inserted for stronger PM flux linkage. In an opposite case, the wider included angle causes higher reluctance, Rm, and weaker PM flux linkage.

In the conventional LPM of the literature, it was assumed that flux density is distributed up to the middle of bridge [21]. However, FEA results shows better agreement with the span inside the magnet cavity suggested in Figure 2. Figure 2 depicts flux density and flux lines over one pole of single layered magnet cavity in FEA.

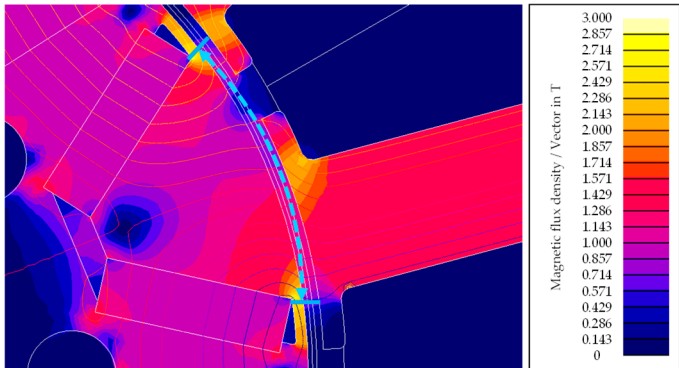

**Figure 2.** Flux density and flux lines over one pole of magnet at no load from finite element analysis (FEA).

*2.2. D-axis Inductance*

As shown in Figure 3, d-axis magnetic path in the core is strongly affected by saturation due to remanent flux if PM is strong enough like rare earth type. Therefore, initial states of core reluctances are already determined before d-axis MMF source is applied. The initial states can be expected from MEC for PM flux linkage mentioned above. Thus, utilizing nonlinear reluctances at previous state, d-axis inductance can be estimated using the Equations (7)–(12).

$$F_{dsn} = \frac{3}{2} \frac{4}{\pi} \frac{N_a k_w}{p^2} I_{dn} \tag{7}$$

$$F_{ds1n} = \frac{1}{\alpha_1 - \alpha_0} \int_{\alpha_0}^{\alpha_1} F_{dsn} \sin(\theta) \, d\theta \tag{8}$$

$$\phi_{dn} = \frac{F_{ds1n}}{r_{g(n-1)} + r_{m(n-1)} + r_{ry1(n-1)} + r_{ry2(n-1)} + r_{teeth(n-1)} + r_{back(n-1)}} \tag{9}$$

$$B_{gn} = B_{g(n-1)} - \phi_{dn} A_g \tag{10}$$

$$\lambda_{dn} = N_a k_w \phi_{dn} \tag{11}$$

$$L_{dn} = \frac{\lambda_{dn}}{I_{dn}} \tag{12}$$

where $I_{dn}$: $n$-th d-axis current, [A], $F_{dsn}$: fundamental amplitude of MMF on airgap at $I_{dn}$, [A], $F_{ds1n}$: constantly distributed MMF value in between $\alpha_0$ and $\alpha_1$, [A], $\phi_{dn}$: flux passing through d-axis path, [Wb], $r_{ry1,2}$: nonlinear reluctances in rotor, [H$^{-1}$], $r_{teeth}$: nonlinear reluctance of teeth in stator, [H$^{-1}$], $r_{back}$: nonlinear reluctance of back iron in stator, [H$^{-1}$], $B_{gn}$: flux density excited by $I_{dn}$, [T], $\lambda_{dn}$: d-axis flux linkage at $I_{dn}$, [Wb], $L_{dn}$: d-axis inductance at $I_{dn}$, [H]. The Equations (7)–(12) describe the formulations of linear and nonlinear reluctances, q-axis MMF source, resulting flux and inductance.

MEC in Figure 3 is constructed by superimposing MEC for d-axis PM flux linkage onto MEC for d-axis inductance including MMF source and nonlinear reluctances in rotor and stator. Moreover, the flow chart in Figure 4 describes the sequence to obtain d-axis inductance based on the previous value of nonlinear reluctances in d-axis path. First, nonlinear reluctances of stator and rotor on d-axis path are substituted into initial values at $I_{dn} = 0$, which is based on air gap flux density obtained in MEC for PM flux linkage. In the next step, at the given d-axis current 1 A$_{rms}$ and resulting MMF source, the flux produced in opposite direction to PM flux linkage can be calculated substituting nonlinear reluctances into Equation (3), and the neutralized flux density of cores must be updated for the next step. This sequence is recurred up to the rated current.

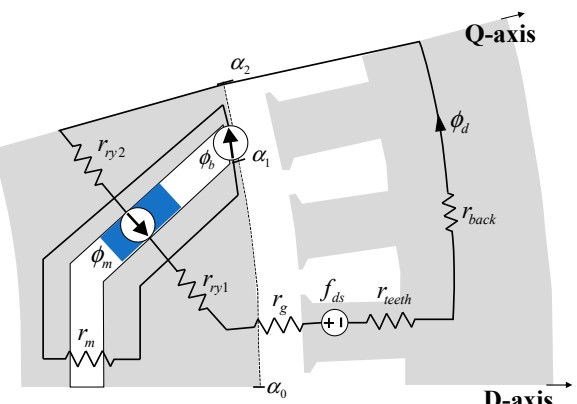

**Figure 3.** MEC for D-axis inductance.

As a result, until neutralizing the saturated core and flux density reaches zero, d-axis inductance gradually increases from $I_{dn} = 0$. This trend will be apparent as more PM is inserted in cavities.

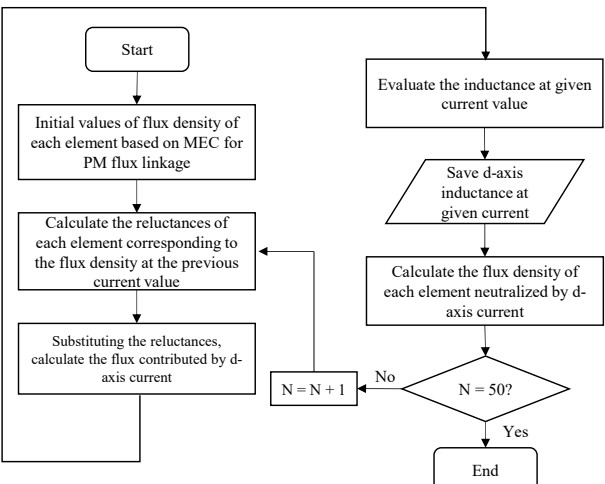

**Figure 4.** Flowchart for D-axis inductance calculation.

### 2.3. Q-Axis Inductance

The solving of q-axis inductance goes in the same way as in d-axis inductance with saturated core in d-axis path due to PM. Q-axis magnetic path is shown in Figure 5. It is turned out in Figure 2 that the entire path is saturated by PM. Therefore, the affection of PM should be considered as in d-axis inductance. Since q-axis MMF does not neutralize d-axis PM flux linkage and is orthogonal to d-axis path, core saturation caused by q-axis flux is simply added to core saturation from d-axis flux. With this fact, Q-axis MEC can be solved using the same sequence in Figure 4.

$$F_{qsn} = \frac{3}{2} \frac{4}{\pi} \frac{N_a k_w}{p^2} I_{qn} \tag{13}$$

$$F_{qskn} = \frac{1}{\alpha_k - \alpha_{k-1}} \int_{\alpha_{k-1}}^{\alpha_k} F_{qsn} \sin(\theta) \, d\theta, \; k = 1, \; 2 \tag{14}$$

$$\Phi_{qkn} = \frac{F_{qskn}}{R_{eqkn}}, \; \lambda_{qn} = N_a k_w \Phi_{qn} \tag{15}$$

$$L_{qn} = \frac{\lambda_{qn}}{I_{qn}} \tag{16}$$

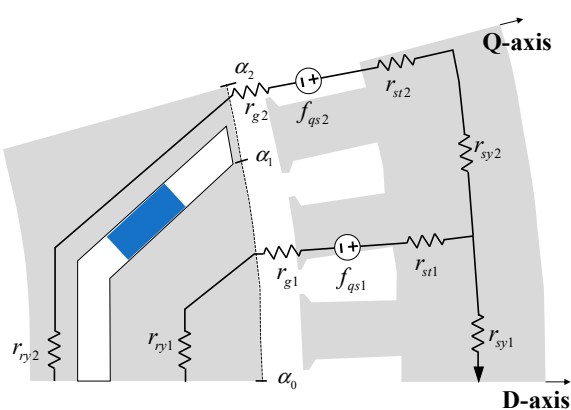

**Figure 5.** MEC for Q-axis inductance.

where $I_{qn}$: $n$-th q-axis current, [A], $F_{qsn}$: fundamental amplitude of MMF on airgap at $I_{qn}$, [A], $F_{qskn}$: values in $k$-th MMF source at $I_{qn}$, [A], $\Phi_{qkn}$: flux passing through $k$-th MMF source at $I_{qn}$, [Wb], $R_{eqkn}$: equivalent reluctance obtained from Kirchhoff's voltage law, [H$^{-1}$], $\lambda_{qn}$: flux linkage at $I_{qn}$, [Wb], $L_{qn}$: q-axis inductance at $I_{qn}$, [H]. Given Equations (13)–(16), relatively low inductance can be estimated provided that magnet cavity is thicker or longer with the narrow included angle. Thicker and longer cavity will make the area of flux path narrow and extend the length so that the rotor yoke can be saturated easily larger amount of PM, and the reluctances grow totally higher. Therefore, if one designs narrow angle of cavity in which to insert more interior PM, it will have low inductance along q-axis current, $I_q$.

### 3. Design of Rotor Shapes

The design for air conditioning compressor required the following specifications: 94 mm of outer diameter, 6000 r/min of base speed with 6 Nm of maximum torque, and 8600 r/min of maximum speed in constant power range. As an initial design to satisfy those requirements, three-phase, 8 poles and 12 slots of stator was designed with tooth-concentrated winding that comes with shorter end-turn winding and resulting small resistance to cut down copper loss. The detailed specifications are tabulated in Table 1.

**Table 1.** Specification of stator and rotor and requirements.

| Parameter | Value | Parameter | Value | Requirement | Value |
|-----------|-------|-----------|-------|-------------|-------|
| Phase | 3 | Stack length | 47 mm | Base torque | 6 Nm |
| Pole | 8 | Diameter of stator | 94 mm | Base speed | 6000 r/min |
| Slot | 12 | Diameter of rotor | 48.8 mm | Max. speed | 8600 r/min |
| Gap | 0.7 mm | Serial turns/phase | 148 turns | Efficiency | 94% |
| Resistance | 0.36 Ω | PM flux density | 1.36 T | | |

Rotors are suggested in Figure 6 as five models according to the included angle and PM volumes. The included angle between two wings of magnet cavity is given from 70 degrees to 110 degrees at an interval of 10 degrees. Each model has the same thickness of PM and more PM is inserted as the angle is smaller. Figure 6 indicates the detailed geometric sizes. Moreover, PM volumes of each model are given in Table 1 with the values of calculated reluctance. By comparing the performance trends according to the angle and the shape of cavities, the cost efficiency on PM volumes will be figured out. In order to investigate about this expectation, FEA results are present in the next section.

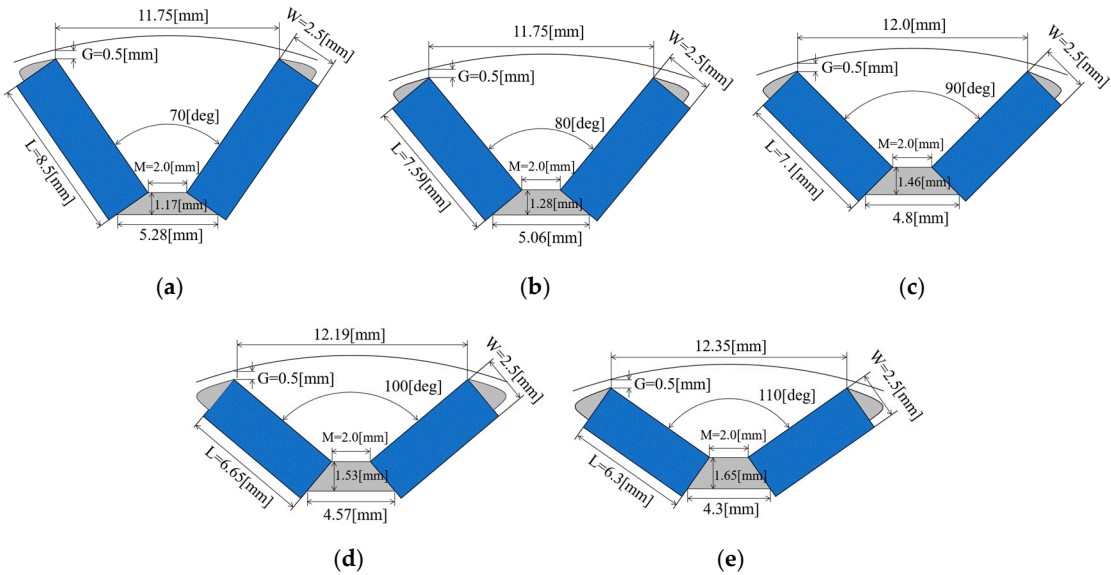

**Figure 6.** The geometric shapes of each model: (**a**) 70 degrees, (**b**) 80 degrees, (**c**) 90 degrees, (**d**) 100 degrees, (**e**) 110 degrees.

## 4. Results of Finite Element Analysis

FEA was performed with a commercial FEA program, Flux2D, to investigate the following characteristics: cogging torque and back-EMF at no-load condition, and d, q-axis inductances and torque production at some load.

### 4.1. Cogging Torque

Cogging torque is produced by the interaction between harmonics of airgap permeance and PM of rotor. Harmonics in airgap permeance is formed from slotted stator and magnet cavities of rotor. Remanent flux distributed by interior PM in rotor stores the energy into airgap in form of magnetic field as in Equation (17). The stored energy is position dependent and its derivative is directly associated with cogging torque. Equation (17) formulates the stored energy in the whole airgap and the association between its derivative and cogging torque [26].

$$W = \frac{1}{2\mu_0} \int_0^{2\pi} B_r{}^2(\theta) \left( \frac{l_c}{l_c + l_g(\theta, \alpha)} \right)^2 d\theta \tag{17}$$

$$T_{cogging} = -\frac{\partial W}{\partial \alpha}$$

where $W$: magnetic field energy stored in gap and cavities, $\theta$: mechanical angular position, $\alpha$: electrical angular position, $T_{cogging}$: cogging torque. Figure 7a shows the curves of cogging torque generated in each model during one cycle of electrical rotation. In Figure 7b, peak and RMS values are given for each model. The decreasing rate in peak and RMS is the steepest between 70 and 80 degrees. Every 10 degrees of variation reduce peak values as much as 42.9%, 26.3%, 24.9% and 1.2% in order from 70 to 110 degree. This decreasing trend continues up to 100 degrees and disappears when change from 100 to 110 degrees. The reductions in PM volumes and the area that stores magnetic field energy are expected as the main reasons.

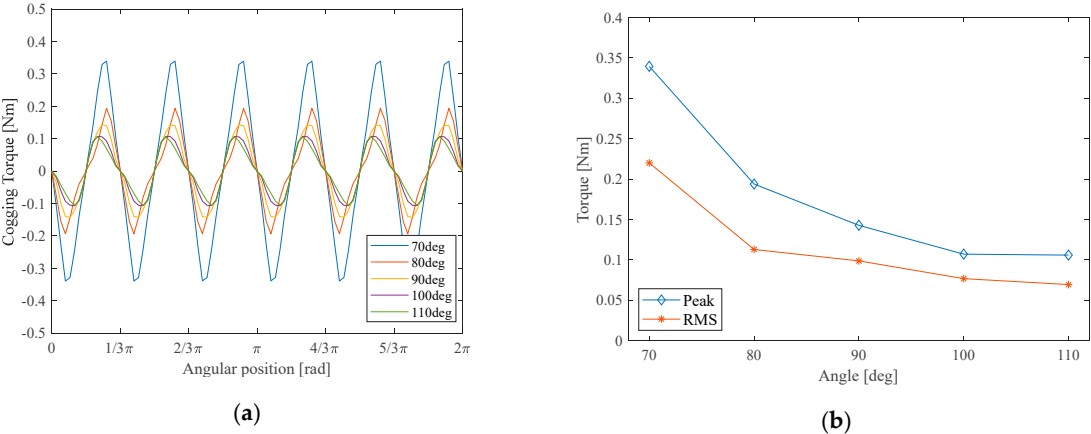

(a)

(b)

**Figure 7.** Cogging torque evaluated from FEA (**a**) curves of cogging torque and (**b**) peak and RMS values of each model.

*4.2. Back-EMF*

From the analysis on back-EMF, its harmonic components and PM flux linkage can be estimated. Hereby, PM flux linkage can be obtained directly from FEA program. Therefore, they are listed in Table 2 for each angle. No-load tests were simulated only at 6000 r/min since other data at different speed are easy to be calculated in linear way; see Equation (18). Evaluated peak values for each model is listed like:

$$E_0 = \omega \lambda_{PM} \tag{18}$$

Figure 8a,b represents the waveform of back-EMF for a cycle and its magnitudes of 1st, 5th, 7th, 9th, and 11th harmonics. In Figure 9, all of values are given in per unit where maximum value among the models at each harmonic order is equal to unity. 1st order harmonics, a fundamental component, decreases at constant rate as the angle becomes wider. Thus, it is envisaged that PM flux linkage also decreases with respect to decline in the fundamental component. On the other hand, 5th, 7th, and 11th harmonics are, with wider angle, more significantly attenuated than 1st order. Only 9th order harmonics tends to increase with the angle in contrary to the other harmonics. However, 9th and 11th order harmonics affect relatively a little while 5th and 7th order of harmonics are the main contributor of torque ripple.

To identify the decreasing rate of THD corresponding to the increase in angle, the values of magnet volumes, PM flux linkage, and THD are represented in Table 2. Compared to magnet volumes and 1st order harmonics, the decreasing rate of THD is apparently higher. Therefore, depending on the requirements, 110-degree model was chosen as the better option.

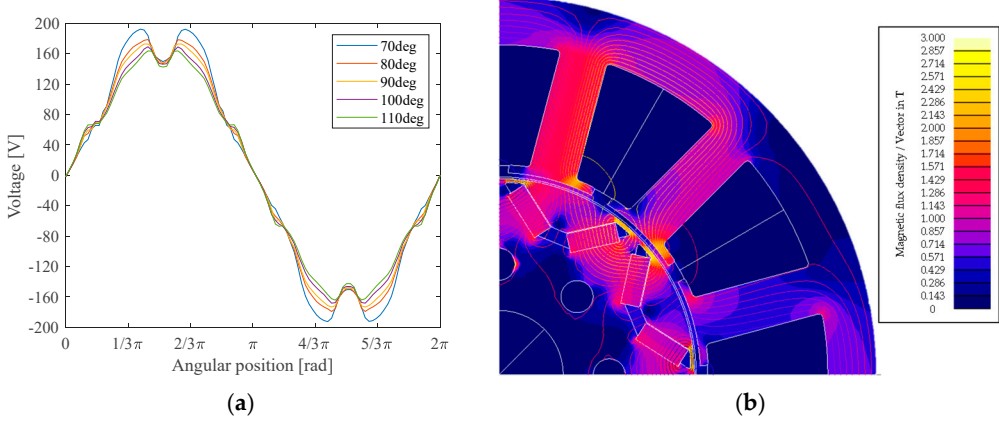

**Figure 8.** Back-EMF at 6000 r/min: (**a**) waveforms of back-EMF (**b**) flux density distribution in FEA.

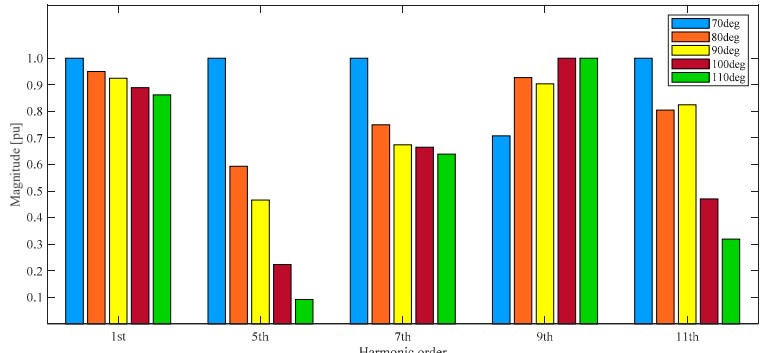

**Figure 9.** Magnitudes of harmonic components in back-EMF.

**Table 2.** PM flux linkage and THD in back-emf.

| Angle [degree] | Magnet Volumes [cm$^3$] | PM Flux Linkage [mWb$_{rms}$] | Back-Emf THD [%] |
|---|---|---|---|
| 70 | 15.98 | 52.3 | 14.42 |
| 80 | 14.31 | 49.67 | 10.02 |
| 90 | 13.16 | 48.34 | 8.74 |
| 100 | 12.53 | 46.49 | 7.01 |
| 110 | 11.80 | 45.06 | 6.65 |

### 4.3. Inductances

Inductances along d and q-axis, respectively, were evaluated while either of d or q-axis current is set during rotation at 6000 r/min. For instance, when d-axis current is set, q-axis current is 0 to prevent the saturation on core by q-axis current. Now that these inductances are to be compared with the expectation from equations in LPM, the same condition must be satisfied.

D and q-axis inductance in Figure 10a both tend to be higher across the overall range as the angle becomes wider. Note that d-axis inductance grows to a certain degree and afterwards becomes constant. The deviation between min and max is observed 28% at 70 degrees and gradually diminished to 4.9% at 110 degrees as getting the angle wider. Min values among the models range from 2.126 to 2.9 mH, which increase by 36%. On the other hand, max value increases by 3.31%. Declines in q-axis inductance are similar in all the models, which are caused by saturation in core. Maximum at 110 degrees is higher by 12.8% than one at 70 degrees, and it is 10.4% in case of minimum. Assuming that an inverter has 100 V$_{dc}$ and 14 A$_{rms}$ and d-axis inductance is constant at the maximum.

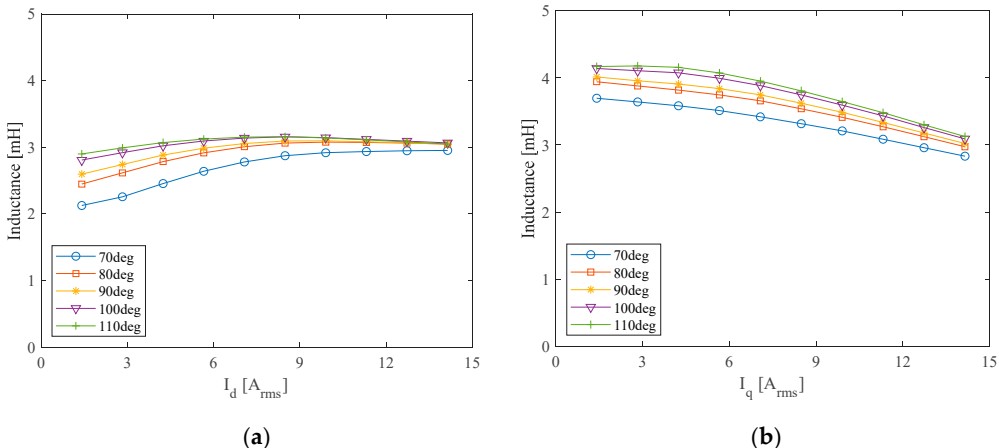

**Figure 10.** D and q-axis inductances: (**a**) d-axis inductance, (**b**) q-axis inductance.

*4.4. Torque*

Under-load test was simulated at rated current, 13.5 A$_{rms}$ and current angles where each model produces maximum torque. Figure 11 is torque curves during one electrical cycle, and more detailed numerical data is tabulated in Table 3, with mean value of torque, torque ripple, and efficiency.

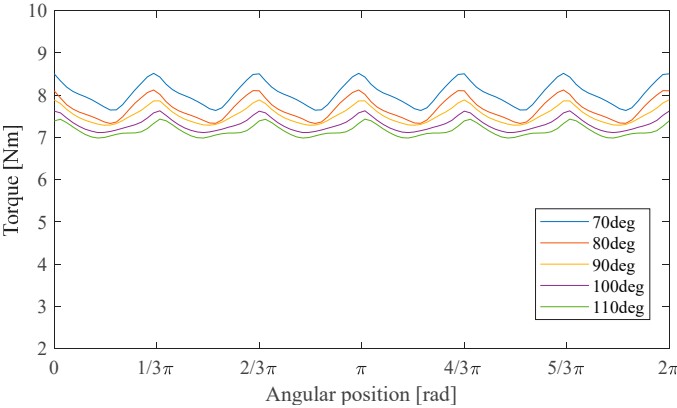

**Figure 11.** Torque curves at 13.5 A$_{rms}$, 6000 r/min for each model.

$$Torque\ ripple\ [\%] = \frac{max - min}{mean} \times 100 \tag{19}$$

DC value of generated torque is the greatest, 8.04 Nm, in the 70-degree model and decreased with wider angles. In case of torque ripple of Equation (19), it tends to be smoother when the angle grows wider from 10.93% to 6.25%. Even though mean value of torque of the 110-degree model is the smallest in Figure 12, 7.16 Nm, torque per magnet volume gradually increases, in other words, getting efficient in cost. The decreasing rate of mean value of torque is 10.95% from 70 to 110 degrees, whereas one of magnet volume is 26.14%. As a consequence, the difference between 70 and 110 degrees in efficiency is 1%, from 94.2% to 93.2%.

These results imply that more reluctance torque is produced as q-axis inductance is reduced since the area of magnet cavity which flux goes through becomes narrow with the wider angle. The attenuation of ripple is originated from the suppression in harmonic components of PM flux linkage shown in Section 4.2. However, the comparison in efficiency is not reasonable in that it is not at the same output power in mechanical operating range.

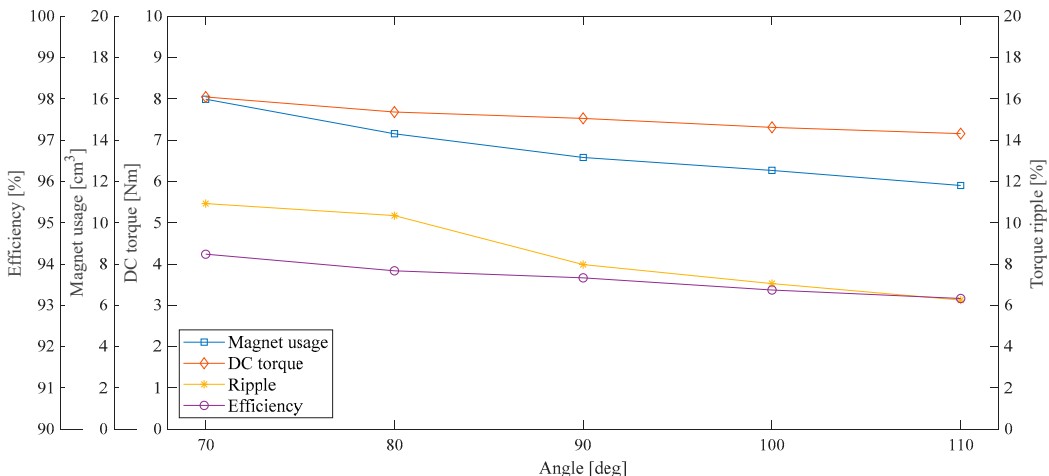

**Figure 12.** Mean value of torque, torque ripple, and efficiency with magnet volume.

**Table 3.** Magnet volumes and resulting performances under load.

| Angle [degree] | Magnet Volumes [cm³] | Mean Value of Torque [Nm] | Torque Ripple [%] | Efficiency [%] |
|---|---|---|---|---|
| 70 | 15.98 | 8.04 | 10.93 | 94.2 |
| 80 | 14.31 | 7.68 | 10.34 | 93.8 |
| 90 | 13.16 | 7.53 | 7.97 | 93.7 |
| 100 | 12.53 | 7.31 | 7.05 | 93.4 |
| 110 | 11.80 | 7.16 | 6.25 | 93.2 |

## 5. Experimental Results

Given the noise caused by torque ripple, EMI, and economic efficiency, the 110-degree model was adopted and manufactured as a prototype to prove the validity of design via experiments. For noise, it is assumed that higher inherent torque ripple causes more acoustic noise according to the literatures [7–12]. Figure 13 is the manufactured prototype of V-shaped IPMSM of which the included angle is 110 degrees. Two sorts of experiments were performed to compare the results from FEA. Back-EMF and MTPA trajectory were identified, which are respectively no-load and under-load tests.

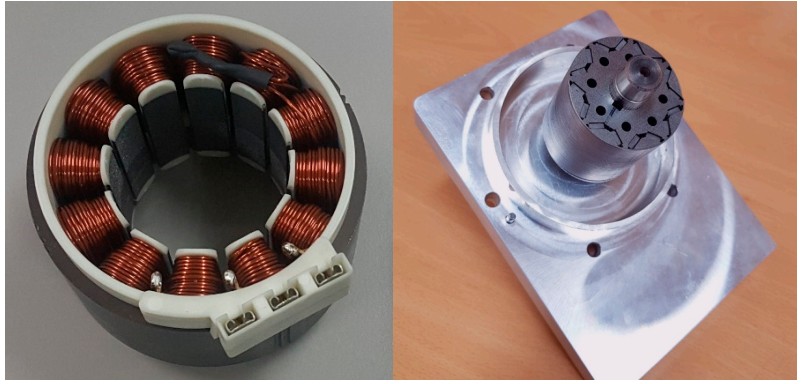

**Figure 13.** Stator and rotor manufactured for experiments.

The experimental testbed is equipped with dynamometer, torque sensor, power analyzer, three-phase inverter and control board with DSP. The entire testbed is described as shown in Figure 14. The dynamometer is connected to the target motor through the torque sensor coupled with both in between them. Three-phase inverter and its control board (DSP) are integrated on the same board

being connected with PC and target motor respectively for a download of firmware and power supply to the target motor. The torque sensor measures while under-load test is ongoing and sends the data to the power analyzer such that power analyzer computes power and efficiency.

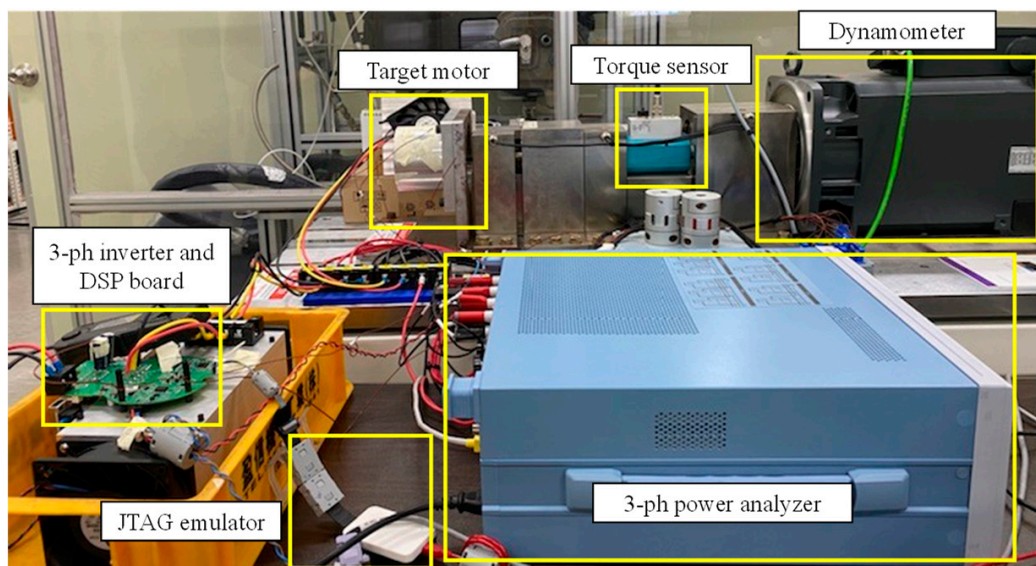

**Figure 14.** The experimental testbed for no-load and under-load tests.

## 5.1. Back-EMF

Dynamometer in Figure 14 provides constant speed control for no-load test of the fabricated motor. Figure 15 are waveforms of phase voltage of U (blue line) and V (purple line) and line to line voltage U to V-phase (green line). Each result was obtained at 6000 r/min and 8000 r/min from power analyzer connected to the terminals of the manufactured motor. Peak values were measured to be 179 $V_{peak}$ for phase voltage and 282 $V_{peak}$ for line to line voltage at 400 Hz (3000 r/min) while at 534 Hz (8000 r/min) 232 $V_{peak}$, 239 $V_{peak}$ for U and V-phase and 372 $V_{peak}$ for line to line are measured. These results are deviated from the results shown in Section 4.2. The reason for this deviation was a gap between the rotor core and interior PM in magnet cavity.

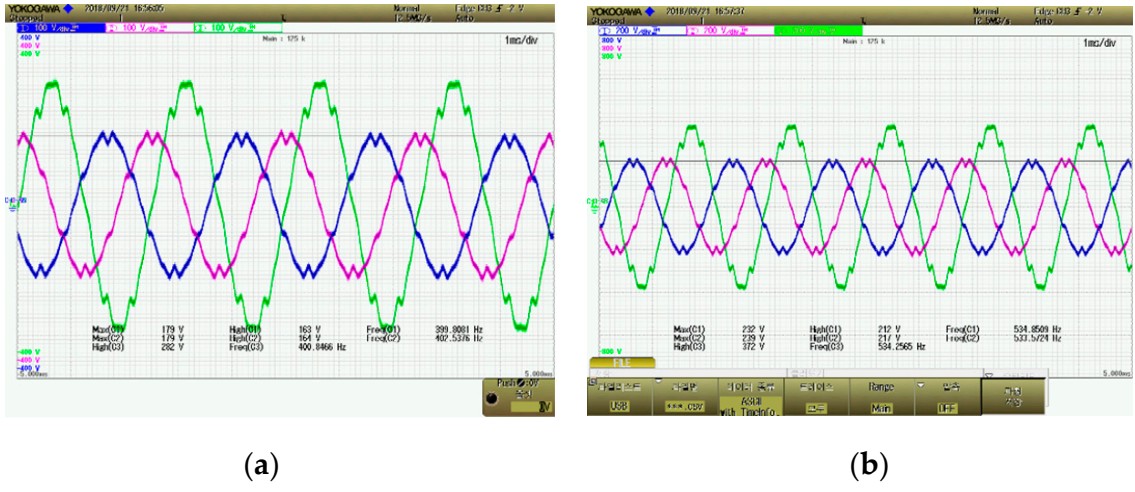

(**a**)　　　　　　　　　　　　　　　　　　　　　(**b**)

**Figure 15.** Measured back-EMF, blue/purple/green: U/V/U to V: (**a**) 6000 r/min and (**b**) 8000 r/min.

Given the gap that appears in magnet cavity, FEA simulation was performed again reflecting the gaps. The waveforms at 6000 r/min are shown in Figure 16a. The agreement in back-EMF is observed if noise in measured back-EMF is ignored. Numerically calculated RMS values are, respectively,

114.57 V$_{rms}$ for FEA and 113.33 V$_{rms}$ for the measured data. RMS values of fundamental component at each speed reference are presented in Figure 16b. The biggest error was found at 4000 r/min, as 2.56%, and mean error at 1000 to 6000 r/min is 2.0%. THD is, respectively, evaluated to be 7.06% from FEA and 5.99% from the measured data. The whole results seem to have good agreements between the fabricated model and the model of FEA simulation.

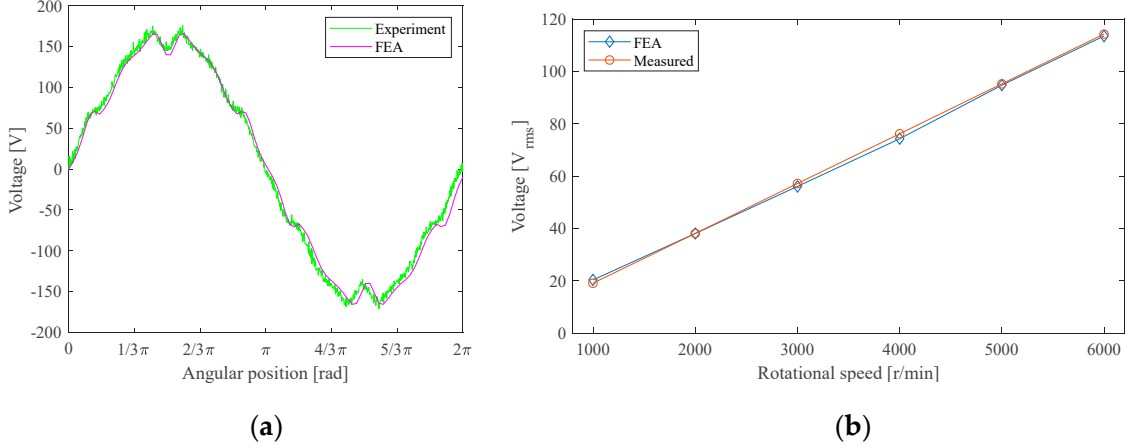

(a)  (b)

**Figure 16.** Comparison between FEA results and measured results on back-EMF: (**a**) waveforms of phase voltage at 6000 r/min and (**b**) RMS values of fundamental magnitude at 1000–6000 r/min.

### 5.2. MTPA Trajectory

To find MTPA trajectory and torque production at current references, under-load test was conducted on the condition where rotational speed of target motor is constantly controlled by dynamometer. After getting the system into steady state, the data of generated torque were measured from torque sensor at given d, q-axis current. The rotational speed was controlled to be 1000 r/min so as to minimize the attenuation in torque production caused by iron loss. Current references range from 1 to 20 A$_{peak}$ (approximately equal to 14 A$_{rms}$) at an interval of 1 A$_{peak}$. Measured current controlled by DSP is given in Figure 17 at 10, 15, and 20 A$_{peak}$.

MTPA trajectories verified from the experiment and FEA are shown in Figure 18a. Figure 18b summarizes the procedure of under-load test to find out the current angle of MTPA. Torque productions at 10, 15, and 20 A$_{peak}$ are evaluated as 3.7 Nm, 5.5 Nm, and 7.18 Nm from FEA and 3.6 Nm, 5.4 Nm, and 6.98 Nm from the measured data. Errors are less than 2.87%. Since FEA did not take consideration of the attenuated torque production due to iron loss, torque production and current angle for maximum torque at each current reference are considered to coincide with design intent in Section 4.3. Efficiency was measured in range from 800 r/min to 8000 r/min and 1 Nm to 6 Nm. The highest efficiency was 94.7% at 3 Nm and 5000 r/min, and the lowest efficiency was 56% at 6 Nm and 800 r/min.

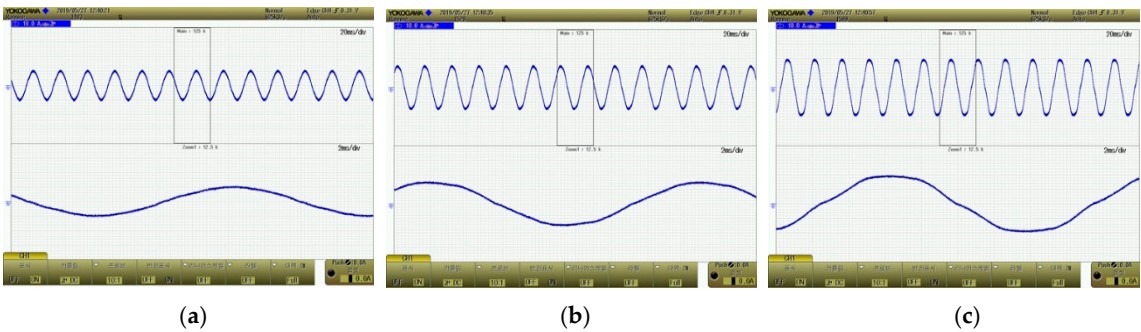

(a)  (b)  (c)

**Figure 17.** Phase current in U-phase at 1000r/min: (**a**) 10 A$_{peak}$, (**b**) 15 A$_{peak}$, (**c**) 20 A$_{peak}$.

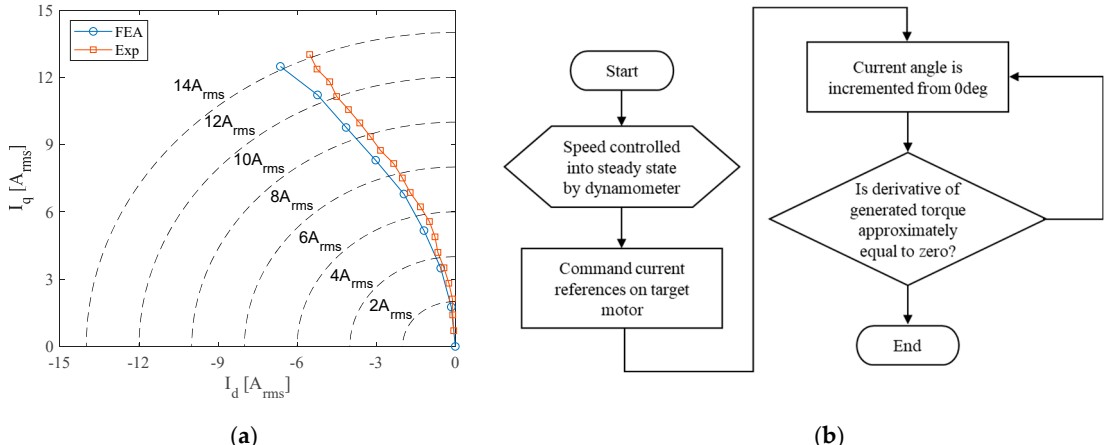

<div align="center">(<b>a</b>)　　　　　　　　　　　　　　　　　(<b>b</b>)</div>

**Figure 18.** (**a**) MTPA trajectories from FEA and experiments and (**b**) flowchart of under-load test.

## 6. Conclusions

This paper investigated the characteristics of no-load and under-load performances according to the included angle in magnet cavity. As a result, back-EMF is lower as its PM is more inserted with narrow angle of magnet cavity. Moreover, the fundamental component of back-EMF increases. However, THD is much more attenuated with the wider angles. It led to smoother torque production, which gives stable operation without severe noise and fluctuation in speed. The results of FEA in Back-EMF and torque are agreed with the prescribed theory based on LPM. Likewise, the trends of inductances along d and q-axis match the expectation based up on LPM. D-axis inductance grows with wider angle, and the 70-degree model tends to increase because d-axis current suppresses d-axis flux from PM, and reluctance saturation gets weaker.

In conclusion, the maximized torque density and efficiency are acquired in the 70-degree model with the largest amount of PM. However, there are disadvantages such as high THD in back-EMF, severe torque ripple, and the highest cost. On the contrary, the 110-degree model has the highest torque per PM volume, smooth torque ripple, and the lowest cost. For the air conditioning compressor in electric vehicles, the 110-degree model is chosen for its cost and low noise characteristics. Experimental tests under load and at no load were conducted and showed the agreement with FEA results.

**Author Contributions:** The literature review and manuscript preparation, as well as the simulations, were carried out by H.J. Experimental results and implementation of the prototype were carried by H.J. and J.B. Final review of manuscript corrections was done by J.B. All authors have read and agreed to the published version of the manuscript.

**Funding:** This research was funded by grant (2018R1D1A3B0704376413) from National Research Foundation of Korea (NRF) and grant (20RTRP-B146053-03) from Railroad Technology Research Program (RTRP) funded by Ministry of Land, Infrastructure and Transport of the Korean government.

**Conflicts of Interest:** The authors declare no conflict of interest.

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
