# Peer review of "Design of Three-Phase V-Shaped Interior Permanent Magnet Synchronous Motor for Air Conditioning Compressor of Electric Vehicle"

_applsci, doi:10.3390/app10248785_

Round 1

Reviewer 1 Report

The authors discuss the importance of noise in IPMSM engines. They do so on lines 45, 50, 51, 282, 335 and 334. Throughout the article, no simulations or experiments are shown to measure the noise produced by the different engine configurations that the authors propose. It would be desirable for the authors to make some to refer this. The authors say in the conclusions:” It is led to the smoother torque production, which gives stable operation without severe noise and fluctuation in speed.”

How do they show it?

Xxxxxx

Line 107: Figure 1. The letters are not visible. The subscripts are very small.

Line 165: Figure 3. The letters are not visible. The subscripts are very small.

Line 189: Figure 5. The letters are not visible. The subscripts are very small.

Line 205: Figure 6. The letters are not visible. The subscripts are very small.

Xxxxxx

Figure 6. The geometric shapes of each model (a) 70degree (b) 80degree (c) 90degree (d) 100degree (e) 110degree.

Better

Figure 6. The geometric shapes of each model (a) 70 degree (b) 80 degree (c) 90 degree (d) 100 degree (e) 110 degree.

Xxxxxx

Line 328:

Figure 18. Phase current in U-phase at 1,000r/min (a) 10Apeak, (b) 15Apeak, (c) 20Apeak.

Better

Figure 18. Phase current in U-phase at 1,000 r/min (a) 10 Apeak, (b) 15 Apeak, (c) 20 Apeak.

Xxxxxx

Line 329: Figure 19(a). The subscripts are very small.

Xxxxxx

From line 119 to line 125; From line 153 to 157; From line 182 to 185; From line 216 to 217

The authors must put the units of the physical quantities.

Xxxxxx

Line 191: 94mm better 94 mm.

Line 192: 6,000r/min better  6,000 r/min. 8,000r/min better  8,000 r/min.

Xxxxxx

Table 1. Columns VALUE: 47mm; 94mm; 48.8mm; 148turns; 1.36T; 6Nm; 6,000r/min; 8,600r/min.

Better

Table 1. Columns VALUE: 47 mm; 94 mm; 48.8 mm; 148 turns; 1.36 T; 6 Nm; 6,000 r/min; 8,600 /min.

Xxxxxx

Line 220: 10degree better 10 degree.

Line 221: 110degree better 110 degree.

Line 227: 6,000r/min better 6,000 r/min.

Line 246: 110degree better 110 degree.

Line 249: 6,000r/min better 6,000 r/min.

Line 254: 70degree better 70 degree.

Line 255: 110degree better 110 degree.

Line 255: 2.9mH better 2.9mH.

Line 258: 110degree better 110 degree. 70degree better 70 degree.

Line 259: 100Vdc better 100 Vdc. 14Arms better 14 Arms.

Line 262: 13.5Arms better 13.5 Arms.

Line 267: 80.4Nm better 80.4 Nm. 70degree better 70 degree.

Line 269: 110degree better 110 degree. 7.16Nm better 7.16 Nm.

Line 271: 110degree better 110 degree.

Line 272: 70degree better 70 degree.

Line 298: 6,000r/min and 8,000r/min better 6,000 r/min and 8,000 r/min.

Line 300: 179Vpeak better 179 Vpeak. 282Vpeak better 282 Vpeak. 400Hz better 400 Hz. 3,000r/min better 3,000 r/min. 534Hz better 534 Hz.

Line 301: 8,000r/min better 8,000 r/min. 232Vpeak better 232 Vpeak.  239Vpeak better 239 Vpeak.  372Vpeak better 372 Vpeak.

Line 305: 6,000r/min better 6,000 r/min.

Line 307: 114.57Vpeak better 114.57 Vpeak.  113.33Vpeak better 113.33 Vpeak.

Line 309: 4,000r/min better 4,000 r/min. 6,000r/min better 6,000 r/min.

Line 318: 20Apeak better 20 Apeak.  14Apeak better 14 Apeak. 1Apeak better 1 Apeak.

Line 319: 20Apeak better 20 Apeak.

Line 322: 20Apeak better 20 Apeak. 3.7Nm better 3.7 Nm. 5.5Nm better 5.5 Nm.  7.18Nm better 7.18 Nm.   3.6Nm better 3.6 Nm.   5.4Nm better 5.4 Nm.

Line 323: 6.98Nm better 6.98 Nm.

Line 326: 800r/min better 800 r/min. 8,000r/min better 8,000 r/min. 1Nm better 1 Nm. 6Nm better 6 Nm.

Line 327: 3Nm better 3 Nm. 5,000r/min better 5,000 r/min. 6Nm better 6 Nm. 800r/min better 800 r/min.

Line 338: 70degree better 70 degree.

Line 340: 70degree better 70 degree.

Line 342: 110degree better 110 degree.

Line 344: 110degree better 110 degree.

Author Response

The authors discuss the importance of noise in IPMSM engines. They do so on lines 45, 50, 51, 282, 335 and 334. Throughout the article, no simulations or experiments are shown to measure the noise produced by the different engine configurations that the authors propose. It would be desirable for the authors to make some to refer this. The authors say in the conclusions:” It is led to the smoother torque production, which gives stable operation without severe noise and fluctuation in speed.”

How do they show it?

Answer: Above all, i appreciate for your thorough review on our paper. The measurement of actual acoustic noise is not convenient while in performance test since inherent torque ripple is compensated by speed controller. And, as you may know, the noise due to torque ripple is mostly caused from mechanical load connected to the electrical machine. For instance, in case where a gear is connected to the electrical machine, a periodical acceleration and deceleration due to torque ripple causes frequent frictions, collision and backlash. From this point, it is not possible to measure the predictive sound noise level with the instruments in our laboratory. Therefore, in this paper, we assumed that lower inherent torque ripple of the electrical machine prevents the audible noise in mechanical load to be connected.

The association between torque ripple and acoustic noise is dealt with in the added literatures [21-26]. Many papers in addition to the literatures [21-24] suggests control methods or system design methods for reduction of torque ripple in various industry applications to achieve lower acoustic noise. Furthermore, the other literatures [25, 26] shows the relationship between torque ripple and acoustic noise. 110 degree model has the lowest acoustic noise with the lowest inherent torque ripple among 5 models. To clarify this for readers, we have inserted the following sentence.

Line 52: “Thus, in the various industry applications, torque ripple should be suppressed for low acoustic noise via dedicated control methods or initial system design [21-24].”

Line 291: “For noise, it is assumed that higher inherent torque ripple causes more acoustic noise according to the literatures [21-26].”

I sincerely appreciate for your review again.

Minor revisions

1

Line 107: Figure 1. The letters are not visible. The subscripts are very small.

Line 165: Figure 3. The letters are not visible. The subscripts are very small.

Line 189: Figure 5. The letters are not visible. The subscripts are very small.

Line 205: Figure 6. The letters are not visible. The subscripts are very small.

Answer: Figures have been edited with larger font sizes.

2

Figure 6. The geometric shapes of each model (a) 70degree (b) 80degree (c) 90degree (d) 100degree (e) 110degree.

Better

Figure 6. The geometric shapes of each model (a) 70 degree (b) 80 degree (c) 90 degree (d) 100 degree (e) 110 degree.

Answer: Clearances have been inserted as recommended.

3

Line 328:

Figure 18. Phase current in U-phase at 1,000r/min (a) 10Apeak, (b) 15Apeak, (c) 20Apeak.

Better

Figure 18. Phase current in U-phase at 1,000 r/min (a) 10 Apeak, (b) 15 Apeak, (c) 20 Apeak.

Answer: Clearances have been inserted as you recommended.

4

Line 329: Figure 19(a). The subscripts are very small.

Answer: Figures have been edited with larger font sizes.

5

From line 119 to line 125; From line 153 to 157; From line 182 to 185; From line 216 to 217

The authors must put the units of the physical quantities.

Answer: The units of the physical quantities have been added.

6

Line 191: 94mm better 94 mm.

Line 192: 6,000r/min better  6,000 r/min. 8,000r/min better  8,000 r/min.

Answer: Clearances have been inserted as recommended.

7

Table 1. Columns VALUE: 47mm; 94mm; 48.8mm; 148turns; 1.36T; 6Nm; 6,000r/min; 8,600r/min.

Better

Table 1. Columns VALUE: 47 mm; 94 mm; 48.8 mm; 148 turns; 1.36 T; 6 Nm; 6,000 r/min; 8,600 /min.

Answer: Clearances have been inserted as recommended.

8

Line 220: 10degree better 10 degree.

Line 221: 110degree better 110 degree.

Line 227: 6,000r/min better 6,000 r/min.

Line 246: 110degree better 110 degree.

Line 249: 6,000r/min better 6,000 r/min.

Line 254: 70degree better 70 degree.

Line 255: 110degree better 110 degree.

Line 255: 2.9mH better 2.9mH.

Line 258: 110degree better 110 degree. 70degree better 70 degree.

Line 259: 100Vdc better 100 Vdc. 14Arms better 14 Arms.

Line 262: 13.5Arms better 13.5 Arms.

Line 267: 80.4Nm better 80.4 Nm. 70degree better 70 degree.

Line 269: 110degree better 110 degree. 7.16Nm better 7.16 Nm.

Line 271: 110degree better 110 degree.

Line 272: 70degree better 70 degree.

Line 298: 6,000r/min and 8,000r/min better 6,000 r/min and 8,000 r/min.

Line 300: 179Vpeak better 179 Vpeak. 282Vpeak better 282 Vpeak. 400Hz better 400 Hz. 3,000r/min better 3,000 r/min. 534Hz better 534 Hz.

Line 301: 8,000r/min better 8,000 r/min. 232Vpeak better 232 Vpeak.  239Vpeak better 239 Vpeak.  372Vpeak better 372 Vpeak.

Line 305: 6,000r/min better 6,000 r/min.

Line 307: 114.57Vpeak better 114.57 Vpeak.  113.33Vpeak better 113.33 Vpeak.

Line 309: 4,000r/min better 4,000 r/min. 6,000r/min better 6,000 r/min.

Line 318: 20Apeak better 20 Apeak.  14Apeak better 14 Apeak. 1Apeak better 1 Apeak.

Line 319: 20Apeak better 20 Apeak.

Line 322: 20Apeak better 20 Apeak. 3.7Nm better 3.7 Nm. 5.5Nm better 5.5 Nm.  7.18Nm better 7.18 Nm.   3.6Nm better 3.6 Nm.   5.4Nm better 5.4 Nm.

Line 323: 6.98Nm better 6.98 Nm.

Line 326: 800r/min better 800 r/min. 8,000r/min better 8,000 r/min. 1Nm better 1 Nm. 6Nm better 6 Nm.

Line 327: 3Nm better 3 Nm. 5,000r/min better 5,000 r/min. 6Nm better 6 Nm. 800r/min better 800 r/min.

Line 338: 70degree better 70 degree.

Line 340: 70degree better 70 degree.

Line 342: 110degree better 110 degree.

Line 344: 110degree better 110 degree.

Answer: Clearances have been inserted as recommended.

Reviewer 2 Report

There are no doubts related the interest for the electrical drive of the air conditioning compressor in electrical vehicles, so that the experience on which the paper is based is useful for the designers of a such systems.

Appreciating the content of the paper I think the extension is a little big, as result of the superposition between two two models of the PMSM design - lumped-parameter model and finite element model. Peoples accept that if both analytical model and finite element model are correct the results are not so far each other and near the experimental ones.

Taking into account the exaggerated colloquial style of the paper text, I propose to reconsider the paper form with much more attention for English and for the scientific style which must characterize an MDPI / applied sciences paper. With this occasion some of my suggestions are as follows:

  • in the title: uppercase for three, and synchronous;
  • line 43: offer roughly numerical values in % of the improvement related the efficiency, power density and power factor of the PMSM drive compared with induction machine drive;
  • line 96: the abbreviations in titles YES or NOT ?;
  • line 145: related "nonlinear reluctances", which is the B(H) nonlinear model considered in the lumped-parameter model ?;
  • line 192: r/min or rpm ?
  • line 206: which software was used for FEA ?
  • lines 210, 211: "Cogging torque ....." definition far from the scientific one;
  • line 213: "derivation" of derivative;
  • line 217, Fig. 7(a): this is not the waveform of cogging torque;
  • Fig. 8(a): the waveform of the Back-EMF voltage is a dependence on time;
  • Line 241: "Magnet usage" means Magnet volume ?
  • Line 248: what is the meaning of " one sort of current" ?
  • Line 264: Mean value of torque is not better than "DC torque" ?
  • Line 270: The decrease rate instead "the decline rate" ?
  • Line 340: It is possible a much clear formulation in English than "70degree model has the .... largest amount of PM" ?

Author Response

There are no doubts related the interest for the electrical drive of the air conditioning compressor in electrical vehicles, so that the experience on which the paper is based is useful for the designers of a such systems.

Appreciating the content of the paper I think the extension is a little big, as result of the superposition between two two models of the PMSM design - lumped-parameter model and finite element model. Peoples accept that if both analytical model and finite element model are correct the results are not so far each other and near the experimental ones.

Taking into account the exaggerated colloquial style of the paper text, I propose to reconsider the paper form with much more attention for English and for the scientific style which must characterize an MDPI / applied sciences paper. With this occasion some of my suggestions are as follows:

Answer: Above all, i appreciate for your thorough review on our paper. As you recommended for our paper, the agreements between analytical model and FEA result were verified within 10% error in the previous researches by the several authors: AlfredoVagati, Edward Carl Francis LOVELACE, Jeihoon Baek, Sai Sudheer Reddy Bonthu and Seungdeong Choi, et al from the literatures [10-15, 17, 20]. Even though it is good to give the values from analytical model, it may occupy the large amount of area to explain them in the limited number of pages. And the purpose of lumped parameter model shown here is to logically anticipate the trends of physical parameters and the final performances with insightful models. Because the results of finite element analysis agreed with the anticipation from analytical model, the comparison is not considered to be an essential part.

Once again, i sincerely appreciate for your review.  

Minor revisions

1

In the title: uppercase for three, and synchronous

Answer: Uppercases for ‘three’ and ‘synchronous’ have been changed as shown in “Design of Three-phase V-shaped Interior Permanent Magnet Synchronous Motor for Air Conditioning Compressor of Electric Vehicle”

2

Line 43: offer roughly numerical values in % of the improvement related the efficiency, power density and power factor of the PMSM drive compared with induction machine drive

Answer: According to the added literature [27], interior PMSM(IPMSM) type achieved the highest efficiency in comparison with synchronous reluctance motor(SynRM), surface mounted PMSM(SPMSM) and induction motor(IM) for air compressor in the same size. Throughout the literature, the efficiency of IPMSM is ranged from 93 to 95% while IM achieved just 85 to 87%. Loss in IM is two times larger than one of IPMSM at least. In addition to efficiency, power factor and power density are higher in the other literatures [28, 29].

 So, the following sentences have been added in line 45: “Throughout the literature, the efficiency of IPMSM was ranged from 93 to 95% while IM achieved from 85 to 87% in the same size[27]. In another literature, power factor of PMSM, 0.95, was significantly higher than one of IM, 0.89 and power density of PMSM was improved by 17.3%[28].”

3

Line 96: the abbreviations in titles YES or NOT?

Answer: All the abbreviations in titles have been substituted as in the second section title, “2. Lumped parameter model of V-shaped IPMSM”.

4

Line 145: related "nonlinear reluctances", which is the B(H) nonlinear model considered in the lumped-parameter model?

Answer: nonlinear reluctances are , , ,  in Figure3. To clarify which is nonlinear reluctances, it has been inserted in line 154 to 156 whether the notations are nonlinear or linear reluctances or not.

5

Line 192: r/min or rpm?

Answer: r/min is considered to be more adequate since journal recommends to use ‘r/min’ in SI unit. So, the unit for rotational speed has been replaced with r/min.

6

Line 206: which software was used for FEA?

Answer: The sentence in line 206 was edited like “FEA was performed with a commercial FEA program, Flux2D, to investigate the following characteristics: cogging torque and back-EMF at no-load condition, and d, q-axis inductances and torque production at some load”.

7

Lines 210, 211: "Cogging torque ....." definition far from the scientific one;

Answer: The paragraph for definition of cogging torque has been totally edited like: “Cogging torque is produced by the interaction between harmonics of airgap permeance and PM of rotor. Harmonics in airgap permeance is formed from slotted stator and magnet cavities of rotor. Remanent flux distributed by interior PM in rotor stores the energy into airgap in form of magnetic field as in equation(17). The stored energy is position dependent and its derivative is directly associated with cogging torque. Equation (17) formulates the stored energy in the whole airgap and the association between its derivative and cogging torque.”

8

Line 213: "derivation" of derivative;

Answer: "derivative" has substituted for "derivation"

9

Line 217, Fig. 7(a): this is not the waveform of cogging torque; Fig. 8(a): the waveform of the Back-EMF voltage is a dependence on time;

Answer: ‘waveform’ in Fig 7(a) has been replaced with ‘curves’. In case of Fig 8(a), as you recommended, back-EMF voltage is a dependence on time. Also as you may know, angular position at a certain rotational speed(6,000 r/min) may be equivalently considered to be time. So, actually x-axis of Fig 8(a), angular position in radian from 0 to 2pi, may be replaced with time from 0 to 2.5ms. Therefore, it seems possible that the waveforms in Fig 8(a) can be with angular position and time both. Hereby, FEA results was directly dependent on angular position and the other papers related to this subject also describe in angular position.

As a result, i hope you understand to use angular position instead of time in Fig 8(a).

10

Line 241: "Magnet usage" means Magnet volume?

Answer: "Volume" has substituted for "usage"

11

Line 248: what is the meaning of "one sort of current"?

Answer: "Either of d and q-axis current" has substituted for one sort of current"

12

Line 264: Mean value of torque is not better than "DC torque"?

Answer: Mean value of torque has substituted for "DC torque"

13

Line 270: The decrease rate instead "the decline rate"?

Answer: decreasing rate has substituted for "the decline rate"

14

Line 340: It is possible a much clear formulation in English than "70degree model has the .... largest amount of PM" ?

Answer: the maximized torque density and efficiency are acquired in 70 degree model with the largest amount of PM.

Reviewer 3 Report

The Authors present the design of PM motors for air conditioning compressor. The prototype has been realized and tested. Interesting and exhaustive paper. 

Author Response

The Authors present the design of PM motors for air conditioning compressor. The prototype has been realized and tested. Interesting and exhaustive paper. 

Answer: I sincerely appreciate for your review on our paper.